# KNOWLEDGE FUSION BY EVOLVING WEIGHTS OF LANGUAGE MODELS

## ABSTRACT

The process of fine-tuning pre-trained language models to aid in downstream NLP tasks is a prevalent technique in NLP research. However, in complex training environments characterized by diverse data domains and tasks, fine-tuned models display varying performance outcomes. The fusion of knowledge across individual models plays a pivotal role in enhancing the performance of a single model. This paper examines the approach of integrating multiple models from diverse training scenarios into a unified model. This unified model excels across various data domains and exhibits the ability to generalize well on out-of-domain data. We propose a knowledge fusion method named model evolving inspired by evolutionary algorithms, which does not need additional training or training data. Our approach involves aggregating the weights of language models into a population and subsequently generating offspring models through mutation and crossover operations. Subsequently, we evaluate the performance of these offspring models in comparison with their parents, thus we can retain the models exhibiting superior performance on the development dataset. Notably, our proposed model evolving strategy can be employed in conjunction with existing model merging techniques, such as *fisher-weighted averaging* and *regmean*. Through a series of rigorous evaluation experiments, we provide empirical evidence that our proposed method significantly outperforms previous approaches.

## 1 INTRODUCTION

In natural language processing (NLP) tasks, due to the high training costs of large language models, it is common practice to directly utilize pre-trained language models and fine-tune them for specific task scenarios. This fine-tuning approach often allows us to achieve excellent performance in specific data domains or tasks at a relatively lower cost (Chen et al., 2021). However, the challenge lies in the fact that fine-tuning (Dodge et al., 2020) the same model in different task scenarios may result in performance variations, meaning that the results may not be satisfactory when testing the same model in different contexts. Therefore, our objective is to integrate knowledge from models trained in different scenarios to enhance the model's performance in cross-domain or cross-task scenarios (Wortsman et al., 2022b).

At present, mainstream knowledge fusion methods can be categorized into two primary groups. The first group involves extensive training on large datasets from several tasks to acquire new model parameters with shared representations. One prominent example within this category is multi-task learning. The second group of methods does not require extensive data but relies on the fusion of existing models from specific scenarios. Generally, multi-task learning methods tend to be more effective in improving overall performance. However, they face two major limitations: the requirement for abundant annotated data for all the tasks simultaneously and the relatively complex and time-consuming nature of multi-task learning algorithms during the training phase (Ruder, 2017), particularly when dealing with dataset combinations. In contrast, model merging methods do not require retraining models and do not raise concerns about data privacy. In this paper, we primarily delve into the second category of methods and introduce an innovative model evolution approach inspired by Darwinian evolution theory (Shafiee et al., 2018). Additionally, we compare model evolution with other prevalent knowledge fusion methods and summarize their characteristics in detail in Table 1.

|  | Ensemble | Model Merging | Multitask Learning | Faderated Learning | Model Soups | **Model Evolution** |
|---|---|---|---|---|---|---|
| Retraining | ✗ | ✗ | ✓ | ✓ | ✗ | ✗ |
| High Memory Cost | ✓ | ✗ | ✗ | ✗ | ✗ | ✗ |
| Round(s) | Single | Single | Single | Multiple | Greedy | Greedy |
| Data | No | A Few Examples | Train Datasets | Private | Dev Datasets | Dev Datasets |
| Key Technique | Inference | Matrices Computing | Distribution | Back-Propagation | Search | Evolution |
| Peak GPU Memory | ✗ | ✓ | ✓ | ✓ | ✗ | ✗ |

Table 1: Comparison of different knowledge fusion methods. Dev datasets means development dataset for validation. Round means the number of times the models are edited when implementing a certain knowledge fusion method. The key technique highlights the difference between knowledge fusion methods. Peak GPU memory is considered low for search or evolution step since only inference is needed. We do not include simple weight averaging method into model merging here.

In fact, the problem of model merging can be reformulated as an optimization problem that aims to find the best strategy to merge or combine these models to achieve better results than any individual model alone. For instance, Jin et al. (2022) employed a simple linear regression approach for optimization, while model soups (Wortsman et al., 2022a) implemented a greedy search method. In this paper, we consider the adoption of a more robust evolutionary algorithm for optimization. Evolutionary algorithms offer several advantages, including their outstanding performance in handling complex, high-dimensional, and nonlinear problems, as well as their relative insensitivity to local optima. In current deep learning research, evolutionary algorithms are primarily used in neural architecture search (NAS) (Awad et al., 2020). However, in this paper, we pioneer their application to the optimization problem of language models for knowledge fusion.

Our approach first processes models fine-tuned in different environments as an initial population. We then generate a new population through mutations and recombination among different individuals within the population. Subsequently, we validate the performance of the new population on development environment datasets and preserve elite individuals for updating. After evolving with enough generations, we select individuals with the best performance as the evolved model. An overview of the entire process is illustrated in Figure 1.

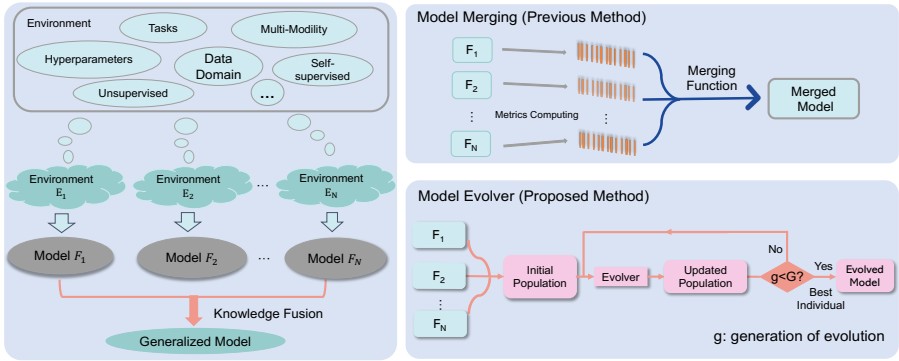

Figure 1: The key step in our proposed model evolution method is to aggregate models $f_{1..N}$ from various environments as a population and updating it through greedy evolutionary rounds. We then compare this to model ensembling and model merging method.

We conduct knowledge fusion experiments across various difficulty levels, employing different types of models, such as RoBERTa and DeBERTa. These experiments encompassed sentiment classification tasks in diverse data domains and benchmark tasks from the GLUE dataset. The experimental results consistently demonstrate that our proposed method effectively enhances performance across all experimental settings. Furthermore, our approach can be synergistically combined with previous model merging methods (*e.g.*, *fisher* (Matena & Raffel, 2022), *regmean* (Jin et al., 2022)), resulting in further improvements in knowledge fusion performance. This combined approach significantly outperforms baseline methods and previous techniques. Notably, our method also exhibits superior generalization performance when applied to data domains beyond the scope of multiple datasets. To summarize, our research contributions include:

- **Innovative model evolution algorithm:** we propose a novel model evolution algorithm (Model Evolution) for knowledge fusion.
- **Improved knowledge fusion performance:** our method consistently enhances knowledge fusion performance across various data domains and tasks.
- **Effective integration with existing model merging methods:** our approach can complement and enhance other model merging techniques.

## 2 RELATED WORKS

### 2.1 KNOWLEDGE FUSION

Numerous studies have shown that aggregating knowledge from multiple datasets can enhance the performance of a single model across various data domains and different tasks (Poth et al., 2021). This approach is also applicable to out-of-domain data (Wang et al., 2020b). Frankle et al. (2020) demonstrated the effectiveness of simple weight averaging in model fusion, exhibiting better performance than pre-training methods. Matena & Raffel (2022) proposed *fisher-weighted averaging* to merge models with different architectures, taking into account the importance of each parameter. Jin et al. (2022) investigated model fusion using regression mean, re-weighting, and linearly combining rows within the weight matrix. Wortsman et al. (2022a) introduced *greedy soup*, a technique to obtain robust results by searching for different average weights from multiple fine-tuned models. Ilharco et al. (2022) proposed the concepts of task vectors to improve pre-trained models on multi-tasks. In addition to these knowledge fusion methods that do not require training, there are many knowledge fusion strategies that require complex training environments. Multi-task learning, as explored by Ruder (2017), improves performance across various tasks by sharing knowledge. Federated learning (Wang et al., 2020a) is a collaborative decentralized privacy-preserving technology designed to overcome the challenges of data silos and data sensitivity. In this paper, we propose model evolution as a knowledge fusion method that is easy to deploy and maintain without the need for retraining.

### 2.2 EVOLUTIONARY ALGORITHMS

Of particular relevance to our work is evolving algorithms (EAs), which provide an alternative path for addressing optimization problems in deep neural networks (DNNs) without relying on gradient information. The fundamental idea behind EAs is to combine the structures and weights of a group of neural networks and continually evolve them in the direction of improved global fitness to enhance model performance. These methods encompass genetic algorithms (Montana et al., 1989), genetic programming (Suganuma et al., 2017), differential evolution (DE) (Pant et al., 2020), and evolutionary strategies (Salimans et al., 2017), among others. Neuro-evolution techniques, such as NEAT (NeuroEvolution of Augmenting Topologies) (Stanley & Miikkulainen, 2002), have demonstrated the ability to design simpler neural network architectures for improved performance, particularly in reinforcement learning tasks. However, it's important to note that EA methods typically perform well on small datasets and small-scale DNNs (Piotrowski, 2014). When applied to large-scale datasets, these methods tend to converge slowly and may even fail to converge (Piotrowski, 2014). In our research, we approach the problem of merging multiple fine-tuned models as an optimization problem. Our proposed model evolution method is motivated by the fact that our problem is neither amenable to traditional gradient-based optimization methods, nor are simple techniques like regression mean sufficient. Therefore, we turn to evolutionary algorithms, which show promise for effectively addressing the model fusion problem.

### 2.3 EXISTING NON-TRAINING-BASED KNOWLEDGE MERGING METHODS

**Fisher-weighted averaging (*fisher*)** examines the importance of each weight $F_i$ associated with each label by computing the norm of the logarithmic likelihood gradient. Specifically, the posterior probabilities of each model are interpreted as gaussian distributions $p(\theta|\theta_i, F_i)$, where the parameters $\theta$ for model $i$ correspond to the Fisher information matrix $F_\theta$. Finally, the fisher information for each parameter is used to perform a weighted average of the parameters, integrating the parameters of different models into a single model.

**Regression mean (*regmean*)** expands the solution of a linear optimization problem to $K$ models where $W_i, i \in \mathcal{K}$, denoted as $W_M = (\sum_i^{i \in \mathcal{K}} X_i^T X_i)^{-1} \sum_i^{i \in \mathcal{K}} (X_i^T X_i W_i)$. Each transformer model $f^{(j)}$ linear layer's corresponding $X_i^{(j)}$ is captured along with per-weights and its input inner product matrix, to compute the merged weights and produced merged model $f_M(x) = W_M^T x$. The scale of $X_i^T X_i$ exhibits substantial variation across different models. Additionally, a control mechanism is applied by multiplying $X_i^T X_i$ by $\alpha = \frac{1}{1+\gamma}$.

**Model soups (*greedy soup*)** Initially, models are ranked based on their development dataset scores. Subsequently, model parameters $\theta_i$ are chosen through a greedy search, and their inclusion in the gradient is determined by comparing the average validation set accuracy before and after their addition. The merged model's parameters can be represented as $\theta_S = \text{average}(\text{ingredients})$.

# 3 EVOLVING WEIGHTS OF MODELS FOR KNOWLEDGE FUSION

This section presents our proposed model evolution strategy. The goal of model evolution is to combine multiple fine-tuned language models into a more powerful single model. We achieve this by simulating the evolution process of a neural network population, as shown in Figure 2. We use the same pre-trained checkpoint and fine-tune it in different environments to create the initial population. As all individuals share the same model architecture, this enables our evolution algorithm to perform mutations and recombinations among individuals within the parameter space.

## 3.1 EVOLUTIONARY STRATEGY

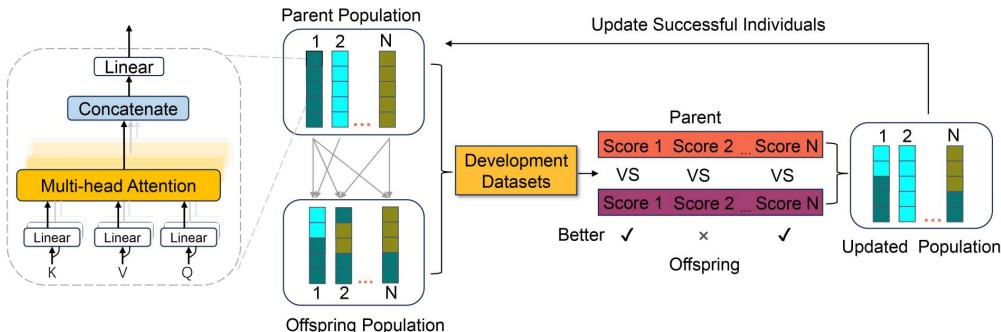

Figure 2: The process of evolving weights of language model.

**Population Initialization** For the optimization problem of model merging, an original set of individuals (population) is initialized. The parameters of each of N models are flattened into a one-dimensional vector, forming a set of candidate solutions. In this way, we obtain a set of candidate individuals represented by $\theta = \theta_i, i = 1, ..., N$. Here, $N$ denotes the size of the population, and $\theta_i = (\theta_{i,1}, \theta_{i,2}, ..., \theta_{i,d})$ represents each candidate individual, where $d$ is the dimension of the solution space.

**Evolution Process** In this phase, we simulate the evolution process of a population of neural networks using the differential evolution algorithm (Pant et al., 2020). Each generation consists of three main steps: mutation, crossover, and updating.

*Mutation:* For each candidate individual $\theta_i$, we randomly select two other candidate individual $\theta_{r_1}$ and $\theta_{r_2}$, where $r_1$ and $r_2$ are two distinct random integers less than or equal to $m$. We use a scaling factor $F$ to adjust the differences between $\theta_{r_1}$ and $\theta_{r_2}$, and then add them to $\theta_i$ to obtain the mutated solution $\theta_i^\star = \theta_i + F \times (\theta_{r_1} - \theta_{r_2})$, where $F$ is used to control the weights of the difference vector in the new parameter set.

*Crossover:* The computation for crossover is as follows:

$$\theta_{i,j}^\star = \begin{cases} \theta_{i,j}^\star & \text{if} \quad \text{rand}(0,1) \leq Cr, \\ \theta_{i,j} & \text{otherwise.} \end{cases} \tag{1}$$

where $Cr$ is the pre-set crossover degree threshold between the new individual and the parent solution, and the setting of the threshold $Cr$ can impact the ratio of elements selected in a mutated solution.

*Updating:* Throughout the process, we convert the offspring population vectors into models and conduct inference to get performance scores for these models on the development dataset. As demonstrated in the equation below, We sequentially evaluate the performance scores of offspring individuals in comparison to their parent one by one. If an offspring performs better, we then replace the corresponding parent individual with it, thereby updating the parent population.

$$\theta_i = \begin{cases} \theta_i^\star & score(\boldsymbol{\theta}_i^{\star g}) > score(\boldsymbol{\theta}_i^g) \\ \theta_i & \text{otherwise.} \end{cases} \tag{2}$$

## 3.2 COMPUTATION EFFICIENCY

**Memory Analysis:** The memory expanse during our model evolution is mainly related to the size of the population: $\sum_{i=1}^{N} d$, where $N$ represents the number of populations, $d$ is the dimension of the model parameter space. Since we avoid inner product matrices computing as in previous model merging methods such as *fisher* and *regmean*, and the parameters is updated mainly through forward propagation of greedy models, the peak GPU memory consumption is consequently lower.

**Time Consumption:** We hereby provide the formula and key definitions required to calculate the runtime. The total evolving time can be calculated as $T = G \times N \times (t_1 + L \times t_2) \approx G \times N \times L \times t_2$, where $t_1 \ll t_2$ in practice. Here, $G$ is the total generations for updating, $N$ is the population size, $t_1$ is the time for mutation and crossover for each individual, $L$ is the number of samples of development datasets, $t_2$ is the time for inference of a sample on one model.

# 4 EXPERIMENTAL SETUP

## 4.1 EVALUATION SETTINGS

**Problems**. We primarily consider the following three main advantages when testing our proposed model evolution method: Firstly, we anticipate that our evolved model $f_M$, created by integrating knowledge from individual models $f_{1..N}$ finetuned on diverse datasets $D_{1..N}$, will have competitive performance across various data sources without necessitating separate models for each domain or task. Then, by evolving different models excelling in various tasks $D_{1..N_t}^t$, we aim to enhance multi-task handling capacity, avoiding the complexity of retraining as in MTL, while enabling cross-task inferencing within a single model. Lastly, our goal is for the evolved model $f_M$ to excel in generalizing to OOD test sets $D_{1..N_o}^o$, thereby enabling it to effectively handle new and unforeseen data from domains or tasks not encountered during training. $D_{1..N}$.

**Datasets**. We chose to employ the GLUE datasets (Wang et al., 2018) as the cornerstone of our investigation into the performance of evolved models. This inquiry encompasses two key dimensions: training for non-i.i.d. partitions and training for disparate tasks. We utilize the emotion classification task as a springboard to explore merged models trained across diverse domains within the same task. For emotion classification, we employ a carefully selected collection of pre-processed datasets obtained from Oberländer & Klinger (2018). Our methodology involves selecting five high-resource datasets for individual model training, alongside the inclusion of five low-resource datasets to evaluate their potential for out-of-domain generalization. Detailed dataset information and additional experimental results are available in Appendix C.

## 4.2 COMPARED METHODS

We mainly compare our proposed model evolution with existing merging methods that do not need retraining, including *simple*, *fisher*, and *regmean*. To gain a better grasp of the advantages of model merging, we show the performance prior to model evolution, the average performance of the population ($Avg. f_{1..N}$) and the best-performing individual ($Best. f_{1..N}$), more details is shown in Appendix C.2. Moreover, we provide the performance for the model trained on a specific task *domain-specific*. We also compare with *model ensembling*, where the logits from predictions are extracted, averaged, and then subjected to the argmax operation. In addition, the *greedy soup* approach requires a held-out dataset for selecting individual models, which is similar to our model evolution method. By greedily identifying new individual models to merge, it can preserve valuable ingredients for weight averaging. However, this method is not suitable for pairwise models merging. Lastly, we use multi-task learning (MTL) as a benchmark for model merging techniques.

## 4.3 EXPERIMENT DETAILS

**Implementation** We make use of Hugging Face's transformer library (Wolf et al., 2019) to access pre-trained language models and conduct fine-tuning. All our models, denoted as $f_i$, follow the same architecture and employ identical pre-trained model weights $\theta$ for initialization, as described in McMahan et al. (2017). Our experiments include various pre-trained models as starting points, such as RoBERTa-base (Liu et al., 2019), the lightweight DistilBert (Khanuja et al., 2021) and

well-established model DeBERTa-large-v3 (He et al., 2021). Besides the models with encoder-only architecture, to establish more generality, we also conduct experiments with encoder-decoder architecture, T5-base-v1.1 (Raffel et al., 2020) and decoder-only architecture, GPT2 (Radford et al., 2019).

**Population Initialization** The initial population for model evolution is created through fine-tuning a model with the same initialization but on different data domains or various tasks. While fine-tuning DistilBERT-base, RoBERTa-base, and DeBERTa-large, we maintained a constant initial learning rate of 1e-5. Throughout our experiments, we consistently utilized the AdamW optimizer with a warm up learning rate during the initial 6% of training. Our model training utilized a batch size of 16 and encompassed 10 epochs for the GLUE task and 30 epochs for the emotion classification task.

## 5 RESULTS

Our primary objective is to evaluate the performance of various training-free knowledge fusion methods (*e.g.*, model merging, greedy soup, ensemble) and compare them to the performance of individual models before fusion. Additionally, we compare these methods with approaches that have higher upper bounds, such as domain-specific techniques and multi-task learning.

We assess the performance dynamics of the model evolution method across a range of scenarios with varying levels of complexity. These scenarios include: (1) performance across different data domains used for fine-tuning individual models. (2) performance across different tasks, when individual models are specialized in only one task. (3) OOD generalization performance on datasets from previously unseen domains. Furthermore, we conduct a series of comparative experiments to analyze the roles of different components in the model evolution process, such as mutation factor, crossover ratio and so on.

### 5.1 MODEL EVOLVING ACROSS DATA DOMAINS

#### 5.1.1 EVOLVING ALL DOMAIN-SPECIFIC MODELS.

We conduct experiments of involving five domain-specific models for emotion classification, and the results are recorded in Table 2. Notably, there is a significant gap between the average performance represented by $Avg. f_{1..N}$ and the best performance indicated by $Best. f_{1..N}$. This is attributed to substantial variations and differences among the $f_{1..N}$ models. Multi-task learning (MTL) achieves performance similar to that of domain-specific models, suggesting that a single model has the capability to acquire knowledge from multiple domains. Additionally, *model soup* approach, which greedily selects fusion objects, leads to some improvements over $Best. f_{1..N}$; However, these improvements are relatively marginal compared to model merging methods.

| Method | Encoder-Decoder T5-base | Encoder-only RoBERTa-base Same / Diff Head Init. | Encoder-only DistilBERT-base Same / Diff Head Init. | Encoder-only DeBERTa-large Same / Diff Head Init. | Decoder-only GPT2 |
|---|---|---|---|---|---|
| Avg. $f_{1..N}$ | 32.07 | 26.08 | 24.55 | 27.68 | 23.35 |
| Best. $f_{1..N}$ | 34.08 | 29.27 | 29.91 | 31.93 | 26.76 |
| Ensemble | 33.95 | 38.77 / 27.73 | 26.51 / 25.43 | 29.88 / 29.27 | 26.82 |
| Greedy Soup | 34.10 | 30.34 | 30.11 | 31.93 | 26.76 |
| Simple | 39.47 | 23.18 | 23.70 | 3.75 | 21.54 |
| Evolver | 41.25 | 33.27 / 30.04 | 28.95 / 26.29 | 23.90 / 21.55 | 23.41 |
| Fisher | 39.12 | 26.09 / 22.43 | 26.39 / 22.61 | 12.83 / 20.42 | 24.93 |
| Fisher_Evolver | 40.36 | 28.41 / 25.71 | 27.63 / 24.75 | 17.22 / **22.95** | 25.66 |
| RegMean | 40.24 | 38.74 / 32.58 | 33.37 / 28.29 | 38.33 / 18.92 | 30.14 |
| **RegMean_Evolver** | **41.83** | **39.87 / 34.28** | **35.67 / 31.11** | **39.58** / 21.79 | **32.26** |
| Domain-Specific | 49.31 | 51.38 | 48.79 | 52.81 | 47.62 |
| MTL | 48.98 | 47.73 | 45.23 | 51.77 | 44.31 |

Table 2: In-domain performance when merging emotion classification models. The initial population are all 5 domain specific models or pairwise models. Simple, Fisher and RegMean are model merging algorithms for comparison. **Bold** numbers indicate the best performance across different model merging algorithms. All the results we reported are averages of trials conducted with 5 different random seeds.

We compare model evolution with three other knowledge merging methods. The basic version of model evolution outperformed *fisher* method and achieved performance that is comparable to *reg-*

*mean* on some tasks. Furthermore, we explore the combined use of model evolution and model merging methods, demonstrating that our approach further enhances existing model merging methods and consistently yields improvements across different models. Also, we demonstrate results with shared and different classification head initialization (Same Head Init/Diff Head Init). It can be observed that *fisher* and *regmean* produce unstable and highly variable results with different initialization, while the performance of the model evolution method is less affected by this factor. Therefore, our proposed model evolution method shows more stable performance when deploying and maintaining a single model across multiple domains.

| Model | Simple | Evolver | Fisher | Fisher_Evolver | RegMean | **RegMean_Evolver** |
|---|---|---|---|---|---|---|
| RoBERTa-base | 37.78 | 39.13 | 37.11 | 40.34 | 46.56 | **46.89** |
| DistillBERT-base | 36.76 | 38.85 | 34.52 | 40.37 | 43.09 | **43.22** |
| T5-base | 38.82 | 40.21 | 38.08 | 41.46 | 47.35 | **47.92** |

Table 3: In-domain performance when merging pairwise domain-specific emotion classification models. All the results we reported are averages of 10 ($\mathcal{C}_5^2$) runs after paring models from a set of 5.

### 5.1.2 EVOLVING PAIRWISE DOMAIN-SPECIFIC MODELS

In addition to the fusion experiment involving all the models, we also consider pairwise domain-specific models for knowledge fusion in the context of the emotion classification task. After pairing models from a set of 5, we conduct 10 ($\mathcal{C}_5^2$) runs with all the same methods described in section 5.1.1. The result of these 10 runs are averaged and recorded in Table 3. We observe clear differences between model evolution and model merging methods, with the *regmean-evolver* achieving the best performance when combining pairwise models finetuned from different domains.

### 5.1.3 EVOLVING MODELS TRAINED ON NON-I.I.D. PARTITIONS.

We adopt synthetic data divisions to simulate non-i.i.d. partitions of the same dataset, across the 8 tasks included in the GLUE benchmark. Given the inconsistency in the performance of *regmean* and *fisher* methods under different seeds, we choose to average the result of eight different random seeds. The results in Figure 3 indicate that the implementation of model evolution outperformed previous methods on all tasks, with clear improvements observed particularly on *cola* and *mnli* datasets. Details of this section are shown in Appendix C.1.

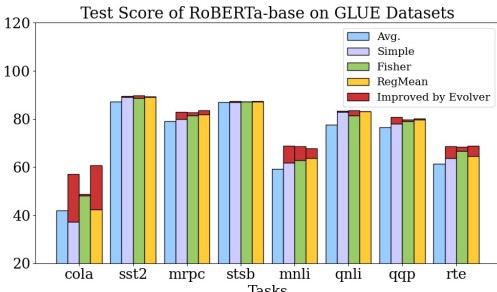
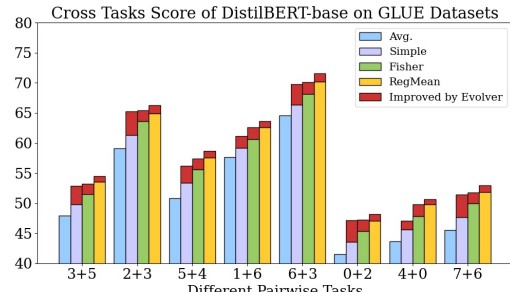

Figure 3: Result of model evolution on non-i.i.d. partitions of GLUE benchmark datasets.

Figure 4: Result of model evolution across different pairwise tasks on GLUE benchmark.

## 5.2 EVOLVING MODELS ACROSS DIFFERENT TASKS

Here we examine the effectiveness of model evolution in merging models finetuned on different tasks. We use the RoBERTa-base model and train individual models with the complete training data for each task in the GLUE benchmark. Following this, we randomly select two task-biased individuals to conduct pairwise model evolution. Specifically, we exclude the parameters in task-specific headers due to their potential dimension variance depending on tasks. We summarize the results of eight different task pairs in Figure 4, which show that our model evolution strategy performs effectively when fusing knowledge from diverse tasks.

## 5.3 Model Evolving for Out-of-Domain Generalization

Regarding Out-of-Domain (OOD) generalization performance, we can obtain the same conclusion as our in-domain experiments, indicating that model evolving leads to improvements in OOD generalization performance, as summarized in Table 7. We notice that in the case of RoBERTa-base and DistilBERT models initialized with different heads, the basic model evolver has outperformed *fisher-evolver* and *regmean-evolver*. A plausible explanation for this is that previous model merging methods may suffer from the negative impact of extremely poor-performing individual models. In the in-domain experiments, this influence is marginal, but it becomes more pronounced in OOD scenarios. In contrast, model evolution possesses an elimination mechanism that effectively removes poorly-performed individual models during the competition process. This ensures that only models with superior performance are retained and merged. Consequently, model evolution can reduce the negative impact of under-performing models on the merged results.

| Method | Encoder-Decoder | Encoder-only | | | Decoder-only |
| | T5-base | RoBERTa-base
Same / Diff Head Init. | DistilBERT-base
Same / Diff Head Init. | DeBERTa-large
Same / Diff Head Init. | GPT2 |
|---|---|---|---|---|---|
| Avg. $f_{1..N}$ | 30.12 | 20.92 | 19.69 | 21.17 | 18.63 |
| Best. $f_{1..N}$ | 37.41 | 29.46 | 29.55 | 31.07 | 27.88 |
| Ensemble | 27.92 | 11.36 / 10.90 | 9.60 / 9.19 | 11.09 / 9.26 | 8.77 |
| Greedy Soup | 15.42 | 13.43 | 15.26 | 4.67 | 11.61 |
| Simple | 38.61 | 11.56 | 13.21 | 0.24 | 10.25 |
| Evolver | 39.26 | 17.53 / **17.16** | 19.02 / **18.42** | 13.33 / 12.78 | 15.87 |
| Fisher | 37.72 | 16.21 / 14.28 | 17.77 / 15.69 | 5.57 / 27.61 | 15.16 |
| Fisher_Evolver | 38.87 | 16.98 / 15.44 | 18.85 / 17.36 | 15.46 / **30.41** | 16.34 |
| RegMean | 39.46 | 21.09 / 14.12 | 18.97 / 16.21 | 15.92 / 4.88 | 20.33 |
| **RegMean_Evolver** | **41.13** | **23.41** / 16.45 | **21.44** / 18.31 | **18.49** / 11.27 | **22.07** |
| MTL | 37.64 | 27.41 | 25.63 | 30.43 | 25.26 |

Table 4: Performance evaluation of model evolution in out-of-domain context.

## 5.4 Ablation Study

**Combination with Other Methods** We demonstrate the evolutionary process when combined with other model fusion methods. From Figure 5, it can be observed that when model evolver is combined with *fisher* or *regmean* method, the upper bounds of the evolutionary approach can be enhanced. Additionally, we present the test results of the evolutionary algorithm on the development dataset. It is evident that as individual models are trained on the development dataset, their performance on the test set gradually improves. This indicates that our model evolution method indeed has the ability to optimize and learn. In addition, *regmean* method requires decreasing the non-diagonal items of the inner product matrices by multiplying a scalar $\alpha$. Since the effectiveness of the *regmean-evolver* method can be influenced by the hyperparameter $\alpha$, we also test the performance of *regmean-evolver* under different $\alpha$ parameters, as shown in Figure 6.

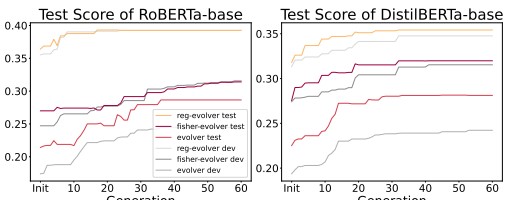
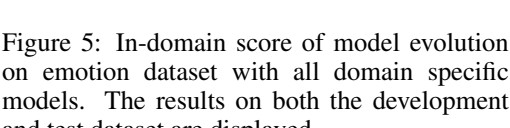
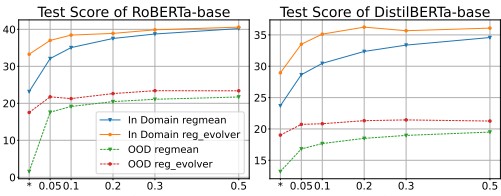

Figure 5: In-domain score of model evolution on emotion dataset with all domain specific models. The results on both the development and test dataset are displayed.

Figure 6: The improvement of *regmean-evolver* with different scale $\alpha$ on emotion dataset when evolving all domain specific models. $*$ means the result of *simple* and *evolver*.

**Mutation and Crossover** We also test the impact of different values of scale factor *F* for mutation and crossover ratio *Cr*, as shown in Figure 7. Due to the inherent randomness in the search process of evolutionary algorithms, we conducted each experiment using four different random seeds and

then calculated the average results. In general, the study findings indicate that the performance of the evolutionary algorithm improves as the parameters $F$ or $Cr$ increase until reaching 0.5. However, when $F$ or $Cr$ exceeds 0.5, there is minimal improvement in performance, and no clear pattern is observed. Therefore, in all experiments conducted in this paper, we have consistently used $F = 0.5$ and $Cr = 0.5$. Furthermore, it is worth noting that the *regmean-evolver* algorithm exhibits a faster convergence rate compared to the simple evolver, typically converging in approximately 20 generations.

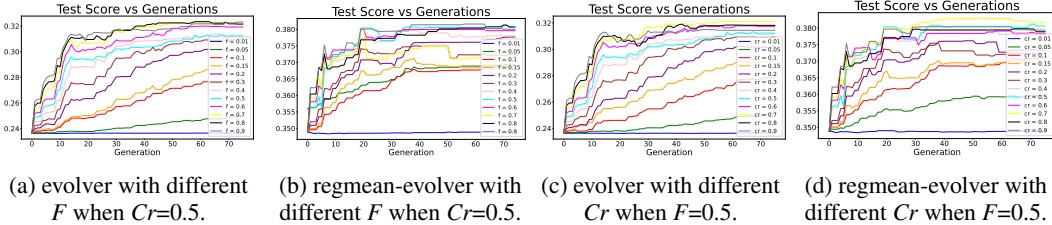

(a) evolver with different $F$ when $Cr$=0.5.

(b) regmean-evolver with different $F$ when $Cr$=0.5.

(c) evolver with different $Cr$ when $F$=0.5.

(d) regmean-evolver with different $Cr$ when $F$=0.5.

Figure 7: Result of RoBERTa-base when evolving all domain specific models on emotion datasets with different with different scale factor $F$ and crossover ratio $Cr$.

## 5.5 ANALYSIS

In this section, we analyze the advantages and limitations of model evolution.

**Advantages.** Actually, the advantages of model evolution are quite prominent. (1) Model evolution can leverage the benefits of a larger population size without being significantly affected by individuals with extremely poor performance. This advantage is a result of the survival of the fittest mechanism in the model evolution process. (2) Model evolution method can effectively maintain low peak GPU memory usage. This is primarily attributed to its sequential forward inference of individual models, as opposed to previous model merging techniques that require additional GPU memory for computing inner product matrices in the model parameter space. This advantage significantly reduces GPU memory consumption and extends the range of feasible solutions for large-scale language models.

**Limitation.** The limitations of the model evolution method can be summarized in three main aspects: (1) the necessity for a high-quality development dataset with consideration for data privacy, (2) the requirement for a cautious selection of hyperparameters $F$ and $Cr$, (for a sensitivity analysis concerning the quality and quantity of the development dataset, please refer to Appendix B) (3) and the significance of conducting further theoretical analysis of evolutionary algorithm principles.

## 6 CONCLUSIONS AND FUTURE WORK

We introduce a novel knowledge fusion method, called model evolution, inspired by evolutionary algorithms. This approach significantly boosts the performance of model merging in diverse NLP contexts. Model evolution stands out by aggregating model weights into a population and updating it with superior offspring models, all without requiring extra training data. Our extensive experiments validate its superiority over previous techniques.

Future research offers several promising directions. Firstly, exploring advanced optimization strategies within evolutionary algorithms, including adaptive approaches and hyperparameter selection based on historical performance, holds the potential for enhancing the method's effectiveness. Secondly, extending knowledge fusion to a more complex training environment by considering hyperparameters, multimodal and exploring different training methods like unsupervised or supervised learning can provide a comprehensive understanding of its applicability. Thirdly, arithmetic operations in Ilharco et al. (2022) for model edit can be analyzed. Lastly, evaluating the approach on larger language models can provide insights into its scalability.

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

## A  EXPLANATION OF MODEL EVOLUTION

In this section, we provide a more comprehensive expla-
nation of the principles underlying the Differential Evo-
lution algorithm, furthermore, we offer an in-depth eluci-
dation of the mutation process, providing a visual repre-
sentation in Figure 8 to enhance clarity and understand-
ing. Overall, the fundamental principle of the differen-
tial evolution algorithm involves randomly selecting three
distinct individuals, performing a mutation operation to
create a new candidate solution, using a crossover opera-
tion to refine the solution, and replacing the original solu-
tion if the new one performs better. This iterative process
continues until certain stopping criteria are met.

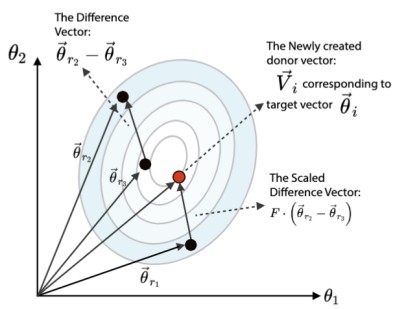

Figure 8: **An illustration of the muta-
tion process in difference evolution.**

To better illustrate our model evolution method, we have
created a flowchart as shown in algorithm 1. The details
of combining our proposed method with other models are
provided in Step 3. The approach involves calculating an
overall score by using model merging on the mutated individuals along with the non-mutated indi-
viduals in the current evolution. In contrast, the simple evolver determines the success of mutation
based on the score of individual entities, while the combined approach assesses it based on the score
of the entire population after individual mutation.

---

**Algorithm 1** Model Evolution

---

1: **Step 1 - Initializing the Population**
2: Initialize population $\Theta$         ▷ A population of candidate solutions
3: $generation \leftarrow 0$         ▷ Initialize generation counter
4: $converged \leftarrow$ **False**         ▷ Convergence flag
5: **while** not converged **do**
6:     **Step 2 - Evolution Process: Mutation and Recombination**
7:     **for** each candidate solution $\theta_i$ in $\Theta$ **do**
8:         Randomly select $\theta_{r1}$ and $\theta_{r2}$         ▷ Select random solutions
9:         $F \leftarrow$ Random scaling factor         ▷ Control parameter for mutation
10:         $Cr \leftarrow$ Random crossover rate         ▷ Control parameter for recombination
11:         Compute mutated solution $\theta_i^\star$ using $\theta_i$, $\theta_{r1}$, $\theta_{r2}$, and $F$
12:         Perform recombination of $\theta_i^\star$ based on $Cr$ and $\theta_i$
13:         **Step 3 - Model Inference**
14:         **if** not combined with other model merging methods **then**
15:             Evaluate the performance of $\theta_i^\star$ on development data
16:         **else**
17:             Merging $\theta_i^\star$ with other models
18:             Evaluate the performance of merged model on development data
19:         **end if**
20:     **end for**
21:     **Step 4 - Updating the Population**
22:     $converged \leftarrow$ **True**         ▷ Assume convergence
23:     **for** each candidate solution $\theta_i$ in $\Theta$ **do**
24:         **if** $\theta_i^\star$ outperforms $\theta_i$ **then**         ▷ Comparing performance
25:             Replace $\theta_i$ with $\theta_i^\star$         ▷ Update population
26:             $converged \leftarrow$ **False**         ▷ Reset convergence flag
27:         **end if**
28:     **end for**
29:     $generation \leftarrow generation + 1$         ▷ Increment generation counter
30: **end while**

---

## B  IMPACT OF DEVELOPMENT DATASET

The availability of development datasets directly impacts the effectiveness of our model evolution approach. However, many publicly available datasets either do not provide development sets or widely use them as test sets. In our case, the development set of the GLUE dataset is used as a test set, so we utilize a small portion of the training dataset (approximately 5%) for model evolution. For non-i.i.d. partition methods, we also use only a subset of the same training data samples. In the case of the unified emotion dataset, we separately extract 10% of data from each of the five high-resource datasets for model evolution, following the same partitioning method as employed in Jin et al. (2022).

For our model evolution approach, the quality of the development dataset can significantly impact performance, making the selection of a high-quality development dataset a crucial consideration. To address this, we conducted experiments on the emotion dataset using different lengths of model evolution methods. We performed experiments with both the simple evolver and regmean evolver, evolving all five domain-specific models. The experimental results are shown in Table 5, indicating that even with a short development dataset, model evolution can still be effective. However, as the length of the development dataset increases, the performance of model evolution tends to improve. Additionally, we included the test scores of simple methods as baselines for comparison.

| Length | None | 1/4 | 1/2 | 1 |
|---|---|---|---|---|
| Evolver | 23.18 | 30.14 | 32.03 | 33.27 |
| Regmean_Evolver | 38.74 | 39.43 | 39.57 | 39.87 |

Table 5: The performance of model evolution with different length of development dataset. *None* means evolver is not conduct and the test score of *simple averaging* and *regmean* method is recorded.

## C  METRICS, DATASET AND TRAINING DETAILS

### C.1  MERGING MODELS TRAINED ON NON-I.I.D. PARTITIONS.

Merging models initially trained on non-i.i.d. partitions of the same dataset is started, which is achieved by simulating synthetic data splits across the 8 tasks within the GLUE benchmark. Each task involves dividing the training data into two partitions, each containing 1,000 training examples with distinct label distributions. Following this, we perform fine-tuning on these two partitions for 8 pairs of individual models and merge each pair of models. The evaluation of these merged models takes place on the official validation sets, which portray a joint distribution of both partitions.

### C.2  METRICS AND COMPARED METHODS

In evaluating merged models trained for non-i.i.d. partitions of the same dataset, we assessed their performance using a unified test set characterized by a joint distribution of all partitions. For merged models trained across different domains or tasks, we measured their performance across individual domains or tasks incorporated into the merger and derived their macro-average. Similarly, when evaluating out-of-domain performance, we computed the macro-average of their performance across the out-of-domain test set.

The performance of individual models involved in merging are reported: (1) the average performance of all individual models (**Avg.** $f_{1..N}$); (2) the performance of the best *single* individual model (**Best.** $f_{1..N}$), as determined by using the

|  | Train | Dev | Test |
|---|---|---|---|
| *In-domain* | | | |
| DialyDialog | 72,085 | 10,298 | 20,596 |
| CrowdFlower | 27,818 | 3,974 | 7,948 |
| TEC | 14,735 | 2,105 | 4,211 |
| Tales-Emotion | 10,339 | 1,477 | 2,955 |
| ISEAR | 5,366 | 766 | 1,534 |
| *Out-of-domain* | | | |
| Emoint | | | 7,102 |
| SSEC | | | 4,868 |
| ElectoralTweets | | | 4,056 |
| GroundedEmotions | | | 2,585 |
| AffectiveText | | | 1,250 |

Table 6: Statistics of emotion classification datasets.

validation set; (3) the performance of the individual models corresponding to the training data set for each test set (**Domain-Specific**).

## C.3 EMOTION CLASSIFICATION

In order to investigate the performance of the sentiment classification task, we selected a diverse and challenging set of datasets. Among them, DailyDialogs (Li et al., 2017), CrowdFlower, TEC (Mohammad, 2012), Tales-Emotion (Alm et al., 2005), and ISEAR (Scherer & Wallbott, 1994) is utilized to train domain-specific model. For acessing OOD generalization performance, we use Emoint (Mohammad & Bravo-Marquez, 2017), SSEC (Schuff et al., 2017), ElectoralTweets (Mohammad et al., 2015), GroundedEmotions (Liu et al., 2017), and AffectiveText (Strapparava & Mihalcea, 2007). For OOD evaluation, we focus exclusively on the fundamental emotions: anger, disgust, fear, joy, sadness, and surprise. A detailed overview of the datasets and statistics is provided in Table 6.

## C.4 GLUE BENCHMARK

In the GLUE dataset experiments, we utilized multiple tasks, including CoLA (Warstadt et al., 2019), SST-2 (Socher et al., 2013), MRPC (Dolan & Brockett, 2005), STS-B (Cer et al., 2017), MNLI (Williams et al., 2018),QNLI (Rajpurkar et al., 2016), QQP, and RTE (Giampiccolo et al., 2007). These tasks cover various natural language understanding problems such as text classification, text similarity, and natural language inference. To assess our merged models, we tested them on the official development sets. We performed experiments by training models on non-i.i.d. partitions, creating various partition scenarios through random sampling. Each partition is uniformly sub-sampled to yield a total of 1,000 training examples per partition.

## C.5 TIME COST

| Initial Population for Evolving | T5-base | RoBERTa-base | DistilBERT-base | DeBERTa-large | GPT2 |
|---|---|---|---|---|---|
| All Domain Specifis Models on Emotion Datasets | 24.5 | 19.2 | 18.7 | 21.3 | 20.1 |
| Pairwise Models on Emotion Datasets | 9.3 | 7.3 | 7.1 | 8.1 | 7.6 |
| None-iid Pairwise Models on GLUE Benchmark | 7.2 | 5.7 | 5.5 | 6.2 | 5.9 |
| Cross Tasks Pairwise Models on GLUE Benchmark | 8.7 | 7.1 | 6.8 | 7.8 | 7.5 |

Table 7: Time cost (in the unit of minutes) of RegMean_Evolver on different experiments with 20 generations. T5-base is tested on single A800 GPU and other models are tested on single A6000 GPU. The time cost is mainly related to the size of model and the length of development dataset when conducting model evolution.

We report the time cost of the scheme of model evolution. T5-base is tested on single NVIDIA A800 80G GPU and other models are tested on single RTX A6000 48G GPU. We find that all task of model evolution can be completed within half an hour, which is very cost-efficient in improving the model performance without further training.

## D INTEGRATION WITH COEFFICIENT SEARCH

| Model | Simple (coefficient search) | Evolver (scale factor search) | Fisher (coefficient search) | Fisher_Evolver (scale factor search) | RegMean (coefficient search) | RegMean_Evolver (scale factor search) |
|---|---|---|---|---|---|---|
| RoBERTa-base | 37.78 (**38.83**) | 39.13 (**39.98**) | 37.11 (**38.96**) | 40.34 (**41.23**) | 46.56 (**46.82**) | 46.89 (**47.03**) |
| DistilBERT-base | 36.76 (**37.63**) | 38.85 (**39.67**) | 34.52 (**37.54**) | 40.37 (**41.31**) | 43.09 (**43.14**) | 43.22 (**43.31**) |
| T5-base | 38.82 (**39.91**) | 40.21 (**41.11**) | 38.08 (**39.22**) | 41.46 (**42.55**) | 47.35 (**47.84**) | 47.92 (**48.06**) |

Table 8: **Coefficient Search Result** when merging pairwise emotion classification models. Simple, Fisher and RegMean are model merging algorithms for comparison. All the results we reported are averages of 10 ($C_5^2$) runs after paring models from a set of 5.

The coefficient search is a promising scheme to improve the model merging performance, by searching the optimal $\alpha$. The proposed model evolution can also be integrated with the coefficient search method by searching the optimal scale factor $f$. We have performed the grid search of $\alpha$ and $f$ with intervals of 0.05 from 0.1 to 0.9. We present the result in Table 8. In the setting of Simple, Fisher and RegMean, the results show that a default version of evolver (with scale factor $f = 0.5, cr = 0.5$) outperforms the coefficient search results. Notably, the integration with scale factor search further

boosts the performance of the evolver, which is worth further investigation. Especially, the crossover ratio $Cr$ in model evolution could also be the subject of coefficient search. Many adaptive schemes are also available in the realm of evolution algorithms, like SADE (Qin & Suganthan, 2005) and SHADE (Tanabe & Fukunaga, 2013), providing a possibility of future algorithmic development.

