# OpenReview forum: "Knowledge Fusion by Evolving Language Models"
_ICLR.cc/2024/Conference — Submitted to ICLR 2024_

### Official Review · Reviewer_zRsU · 2023-10-29

**Soundness:** 3 good
**Presentation:** 2 fair
**Contribution:** 3 good
**Rating:** 3
**Confidence:** 4

**Summary:**

The paper proposes a knowledge fusion method inspired by evolutionary algorithms, which doesn't require additional training or data. The method involves aggregating language model weights into a population and generating offspring models through mutation and crossover operations.The proposed method outperforms previous approaches on various settings.

**Strengths:**

+ The paper introduces a novel knowledge fusion method inspired by evolutionary algorithms. This approach doesn't require additional training or data, making it unique in the realm of NLP research.
+ The paper conducts rigorous evaluation experiments, providing empirical evidence that their proposed method significantly outperforms previous approaches.

**Weaknesses:**

+ The motivation is unclear. Authors mentioned in the introduction part that multi-task learning is one of the two main-stream knowledge fusion methods but it suffers from high annotation cost and complex algorithm. However, I can't see why multi-task learning would be more data-hungry than first training individual models on each dataset and then merging them into a single one. As for the second limitation, still, I am not convinced that multi-task learning would be more complex than existing model merging algorithms.  Moreover, the author postulate that model merging is an optimization problem. However, there seems to be no further explanation, e.g., what is the goal of the optimization "problem"?
+ Table 1 is Confusing. What do you mean by "round" and "key step"? More explanation is in need.
+ The structure of the submission still have rooms for improvement. For example, 3.1 is not necessarily a preliminary or premise to understand the method. Therefore I would suggest moving this part to related work. Besides, the experiments in Section 5.4 Ablation Study are not ablation experiments but hyper-parameter analysis, strictly speaking.
+ Missing related work. The task arithmetic  should be discussed, with EDITING MODELS WITH TASK ARITHMETIC as a representative example.

**Questions:**

See the weakness above.

---

> ### Author Response · Authors · 2023-11-19
> **First response to Reviewer zRsU**
>
> We first thank the reviewer for the detailed comments and insightful feedback, which will help us improve the quality of this paper. We address the concerns of weaknesses (W) and questions (Q) below:
>
> W1: Thank you for asking for more clarity about the motivation. We rephrase the sentences in the manuscript to emphasize the main motivations and provide the responses to the Reviewer concerns below:
>
> Multi-task learning necessitates abundant annotated data for all tasks concurrently during training, unlike model merging algorithms, which do not demand simultaneous data availability for each task. Regarding complexity, model merging doesn't mandate the costly retraining of models for all tasks simultaneously. Additionally, model merging can be viewed as an optimization aiming to discover the optimal strategy for merging multiple models to outperform any single model in isolation.
>
> In practice, our proposed method, and the related work like RegMean perform well in the situation where the initial models are prepared, so usually the details how they are finetuned are not the bottleneck of the method.
>
> W2: Thank you for the clarification. We have added the explanation in the caption of Table 1. Round means the number of times the models are edited when implementing a certain knowledge fusion method. The key step is renamed to key technique to highlight the difference between knowledge fusion methods.
>
> W3: Thank you for the suggestions. We have restructured sections 2 and 3 as suggested to move the previous preliminary to related work. For section 5.4, the second part is indeed hyperparameter analysis. However, the first part is an ablation study that demonstrates the proposed model evolution can complement existing model merging methods for enhanced performance.
>
> W4: Thank you for the suggestion. We have added the discussion in the manuscript on the related work [1] in Section 2 and Section 6. In addition, we have cited two important related work about robust finetuning [2] [3] in the manuscript for further reference.
>
> [1] Ilharco, G., Ribeiro, M. T., Wortsman, M., Gururangan, S., Schmidt, L., Hajishirzi, H., & Farhadi, A. (2022). Editing models with task arithmetic. arXiv preprint arXiv:2212.04089.
>
> [2] Jesse Dodge, Gabriel Ilharco, Roy Schwartz, Ali Farhadi, Hannaneh Hajishirzi, and Noah Smith. Fine-tuning pretrained language models: Weight initializations, data orders, and early stopping. arXiv preprint arXiv:2002.06305, 2020.
>
> [3] Mitchell Wortsman, Gabriel Ilharco, Jong Wook Kim, Mike Li, Simon Kornblith, Rebecca Roelofs, Raphael Gontijo Lopes, Hannaneh Hajishirzi, Ali Farhadi, Hongseok Namkoong, et al. Robust fine-tuning of zero-shot models. In Proceedings of the IEEE/CVF Conference on Computer Vision and Pattern Recognition, pp. 7959–7971, 2022b.
>
> We thank the Reviewer again for the useful comments. We have revised the manuscript according to the Reviewer’s suggestion and response to each comment provided in the Weakness section above. We hope that our rebuttal aligns with the reviewer’s expectations, and we hope that the Reviewer can consider possibly giving a higher rating. Thanks.

---

### Official Review · Reviewer_zN3R · 2023-10-30

**Soundness:** 2 fair
**Presentation:** 2 fair
**Contribution:** 2 fair
**Rating:** 5
**Confidence:** 5

**Summary:**

In this paper, the authors introduce a novel approach to knowledge fusion called model evolution, which draws inspiration from evolutionary algorithms. This technique involves pooling the weights of various language models into a population and then generating new models through mutation and crossover operations. The performance of these new models is subsequently assessed, and those exhibiting superior performance are retained. This approach not only attains results comparable to prior merging methods but can also be used in conjunction with them to achieve even better performance.

**Strengths:**

The motivation is clear and the research question is very interesting: The fusion of knowledge and strengths from individual language models is crucial as it can enhance the performance of a single model with minimal computational and time costs. The author has devised a novel method utilizing evolutionary algorithms for model merging.

**Weaknesses:**

1. The paper suggests the direct application of existing evolutionary algorithms for knowledge fusion, which is of limited novelty, yet it lacks an explanation for why evolutionary algorithms can ensure convergence to an optimal result. Furthermore, there is a significant concern regarding the substantial search cost incurred during the evolution process.

2. Absence of experiments involving natural language generation and other model architectures (e.g., encoder-decoder or decoder-only): All experiments are based on the encoder-only model for natural language understanding tasks. It would be valuable to observe experiments using encoder-decoder or decoder-only models, especially in natural language generation tasks.

3. The presentation of this paper is subpar. In addition to language issues, numerous crucial concepts are not explained clearly.

**Questions:**

1.   What are the implementation details of combining evolver with other model merging methods?
2.   What are the details of Avg. *f*1*..*N, Best. *f*1*..*N, Domain-Specific, and MTL? Especially, what is the difference between Best. *f*1*..*N and Domain-Specific?
3.   Seems that cannot find the details of Section 5.1.3 in Appendix B.
4. It appears that the search cost during the evolutionary process is substantial. While the authors have conducted an analysis of time consumption, I am left wondering about the magnitude of this time cost when compared to other methods. Additionally, I'm interested in understanding how to ensure convergence to an optimal result.

---

> ### Author Response · Authors · 2023-11-19
> **First response to Reviewer zN3R**
>
> Thank you for recognizing our research work. We also thank the reviewer for the detailed comments and insightful feedback, which will help us improve the quality of this paper. We address the concerns of weaknesses (W) and questions (Q) below:
>
> W1: Thank you for requesting more clarity on evolutionary algorithms (EA) and the related computing cost.
>
> Proving convergence in EAs is challenging due to the stochastic and heuristic nature and is usually done using empirical studies and analyses.
>
> Despite that, EAs are adept at exploring complex and high-dimensional search spaces, where traditional optimization or search algorithms might struggle (e.g., learning tasks with complex and non-differentiable objective functions). Hence, EAs have been successfully applied in various fields, including engineering, robotics, finance, biology, and more, solving complex real-world problems where traditional methods fall short. Moreover, EAs are complementary to other techniques and yield improved performance.
>
> The computation cost is discussed in Appendix C.5. We found that the computation cost is reasonable, i.e., the evolution process can be completed within half an hour using either A6000 or A800 GPU. The ability to produce higher-quality solutions in diverse NLP tasks makes evolution process valuable for research and practical applications. Moreover, we demonstrate that combining EA with existing methods yields more efficient solutions in our ablation study (Section 5.4). Hence, continued research can aim at improving EAs' efficiency, scalability, and applicability across a broader range of problem domains.
>
> W2: Thank you very much for this insightful suggestion. We have incorporated additional architectures, T5-base and GPT2, into our experiments, as suggested by Reviewers 1 and 3. The new results have been added to Tables 2 and 4, demonstrating the consistently superior performance of our proposed method compared to existing methods. This reinforces the generalizability of our approach to virous neural architectures.
>
> W3: Thank you for pointing out the issue. We acknowledge the need for a clearer presentation and have revised the manuscript accordingly. Concepts, such as multi-task learning and the evolutionary algorithm for optimization, have been explained in more detail in the Introduction. We also discuss more details about compared method and experiment settings in Appendix C. Also, we provide explanation to the Reviewer’s questions in the following and revised the manuscript accordingly. We believe these suggested changes have improved the overall quality of the presentation.
>
> Q1: Thank you for the question. The implementation details of combining evolver with other model merging methods are established in the computation of the population's scores after mutation and crossover. The combining approach involves calculating an overall score by using model merging on the mutated individuals along with the non-mutated individuals in the current evolution. In contrast, the simple evolver determines the success of mutation based on the score of individual entities, while combining assesses it based on the score of the entire population after individual mutation. We added a paragraph in Appendix A and details are shown in step 3 (model inference) in Algorithm 1. For further reference, the reader could also refer to the source code.
>
> Q2: Thanks for requesting more clarity on description of abbreviations. We now provide more explanation in Appendix C.2 accordingly to enhance readability. The following are the details:
>
> Avg. $f1*..N$ shows the average performance of all individual models. Best. $f1..N$ shows the performance of the best single individual model, as determined by using the validation set. Domain-Specific shows the performance of the individual models corresponding to the training data set for each test set. MTL shows the performance of the model trained with multi-task learning method.
>
> Q3: Thank you for the careful check. The details of Section 5.1.3 are in Appendix C.1 instead of Appendix B. We have corrected the manuscript accordingly.
>
>
>
> Q4: The computational cost of our proposed method has been detailed in Section C.5 of the supplementary materials. Our results indicate that model merging can be completed within an hour with moderate computing power, underscoring the practicality of our approach.
>
> We thank the Reviewer again for the useful comments. We have revised the manuscript according to the Reviewer’s suggestion and response to each comment provided in the Weakness section above. We hope that our rebuttal aligns with the reviewer’s expectations, and we hope that the Reviewer can consider possibly giving a higher rating. Thanks.

---

### Official Review · Reviewer_UukK · 2023-11-03

**Soundness:** 2 fair
**Presentation:** 1 poor
**Contribution:** 2 fair
**Rating:** 3
**Confidence:** 5

**Summary:**

The paper proposes to evolve models with mutation and crossover operations over a set of trained models. The algorithm can be built upon different model merging strategies such as Fisher-weighted averaging and RegMean and improves performance over merged models without mutation.

**Strengths:**

- The idea of applying mutation algorithms to create a merged model out of existing models is inspiring and we see clear performance improvement over counterparts without mutation.
- I appreciate combination of mutation algorithms with different model merging approaches
- Ablation studies and hyperparameter sensitivity analysis in Sec 5.4 is quite useful.

**Weaknesses:**

- In my perspective, the major issue is the presentation of the paper.

I find the design of the diagrams, tables, and experiment setups overly similar to a paper authors cited [1], namely Figure 1, Table 2, Table 3. At first glance, I was very confused because of the similarity; until I realized that the submission indeed proposes novel ideas and present interesting new results.

I believe whether the similarity matters is subjective, as it is inevitable for follow-up studies to apply the same experiment setups. Therefore, I would like to hand over the issue to the Area Chair. At the same time, I hope to hear from authors about any plans to modify layouts of Figure 1, Table 2, Table 3 to avoid potential confusion.

There are also other minor writing issues in the paper, like, citations should not be in parenthesis when the authors are included in a sentence.

- Issue with the evaluation

The authors assumes a setup where a develop set is available to evaluate the performance of merged and individual models. In this case, an intuitive baseline is to tune the coefficient $\alpha$ of models to be merged, Merged = $\alpha$ Model1 + $(1-\alpha)$ Model2 , like as Matena et al. or [2]. Especially in the setup of merging only two models, I don't see a reason how the proposed approach can outperform coefficient search.


 [1] Jin et al. Dataless Knowledge Fusion by Merging Weights of Language Models, ICLR 23

 [2] Ilharco et al. Editing models with task arithmetic, ICLR 23

**Questions:**

- What is the performance of merged models with simple average / regmean / fisher-weighted averaging when you apply coefficient search? Does the proposed approach improve over coefficient search?

---

> ### Author Response · Authors · 2023-11-19
> **First response to Reviewer UukK**
>
> We first thank the reviewer for the detailed comments and insightful feedback, which will help us improve the quality of this paper. We address the concerns of weaknesses (W) and questions (Q) below:
>
> W1: Thank you for the point out the issue. We have re-designed diagrams and tables to avoid potential confusion with paper [1]. Our motivation of taking reference from the experimental setup in the paper [1] is mainly for fair and sufficient comparison with previous methods. The clear comparison could allow us to emphasize the main contribution of our paper that is the newly proposed model evolution algorithm. Furthermore, to further we have used a new experiment setup to compare the performance with coefficient search in Appendix D. This helps further improve the overall novelty of our paper.
>
> We have revised Fig.1. Since the parts on ensembling and model merging are not our contribution, we remove the part of ensembling and clarify that model merging is the previous method and model evolution is our proposed method. In Table 2 and Table 4, we have added the results of models with encoder-decoder architecture and decoder-only architecture. The modified figures and tables are shown in our new version of the paper. Lastly, we have modified a few minor writing issues about citations, i.e., we remove the parenthesis of citation when the author is included in the sentence. We hope that these changes have addressed the Reviewer's concerns about the presentation of our manuscript.
>
> W2/ Q1: Thank you for the insightful suggestion of using coefficient search. We are grateful that the Reviewer raises this essential concern, which could potentially inspire the development of novel algorithm.
>
> We would like to answer this question from both the theory and experiment aspects. Theoretically, a coefficient search can be treated as a method with greedy rounds aiming to obtain better efficiency for different models to merge, but a vanilla coefficient search method does not consider the crossover ratio for a better search result.
>
> We have conducted the experiment to investigate the coefficient search. The coefficient search is a promising scheme to improve the model merging performance, by searching the optimal $\alpha$. We find that the proposed model evolution can also be integrated with the coefficient search method by searching the optimal scale factor $f$. We have performed the grid search of $alpha$ and $f$ with intervals of 0.05 from 0.1 to 0.9. We present the result in Appendix D. In the setting of Simple, Fisher and RegMean, the results show that a default version of evolver (with scale factor $f=0.5, cr=0.5$) outperforms the coefficient search results. Notably, the integration with scale factor search could further boosts the performance of the evolver, which is worth further investigation.
>
> We thank the Reviewer again for the useful comments. We have revised the manuscript according to the Reviewer’s suggestion and response to each comment provided in the Weakness section above. We hope that our rebuttal aligns with the reviewer’s expectations, and we hope that the Reviewer can consider possibly giving a higher rating. Thanks.

---

### Official Review · Reviewer_8A57 · 2023-11-03

**Soundness:** 3 good
**Presentation:** 3 good
**Contribution:** 3 good
**Rating:** 6
**Confidence:** 4

**Summary:**

The paper studies the problem of knowledge fusion across multiple models, which would help with modularity and promises to improve performance on in domain and out of domain tasks.

The presented method is based on evolutionary algorithms, where multiple models are initially trained and then evolved and recombined into new models over multiple rounds. Development sets are needed to guide the evolution process across rounds. The method can be combined with model merging approaches, which in contrast, perform knowledge fusion across a single round.

The experiments are performed on the same experimental setups as in the paper introducing RegMean, including the same setups, data sets and initializations.  Results show that the evolutionary algorithm performs better than other methods like greedy soup and mostly better than Fisher-weighted averaging, albeit usually lower than the best model merging method. However, when combined with model merging approaches like RegMean and Fisher-weighted averaging, it leads to significantly better results than evolution or merging alone.

The novelty of the paper is the experimental results and application of existing approaches in evolutionary algorithms to this problem of knowledge fusion. The experimental setup and algorithm are not novel.

**Strengths:**

The results are positive and consistent.

Solid evaluation setup.

The interpretation of the results is quite intuitive regarding removing the models with low performance, which was observed also as a weakness in past work.

Sensitivity analyses conducted.

**Weaknesses:**

The limitation of having enough development data for each domain for the evolution could be a strong constraint for the data privacy setup, which could limit the applicability of the method and was an important selling point for model merging. This just needs to be highlighted better in the paper.

It would have been good to test the approach also with encoder-decoder (like in the RegMean paper) or with decoder-only architectures, to establish more generality.

**Questions:**

The paper should be checked for typos

e.g. Table 1  Faderated > Federated

---

> ### Author Response · Authors · 2023-11-19
> **First response to Reviewer 8A57**
>
> We first thank the Reviewer for recognizing the contribution of our work and the insightful feedback, which will help us improve the quality of this paper. We address the concerns of weaknesses (W) and questions (Q) below:
>
> W1: Thank you for the suggestion. We have incorporated the discussion and highlighted the main selling point of model merging for data privacy, as our limitation in the revised manuscript in Section 5. Our method can incorporate additional consideration for data privacy in the development data as future work. Furthermore, we added more discussion about development data in Appendix B.
>
> W2: Thank you very much for this insightful suggestion. We have incorporated additional architectures, T5 and GPT2, into our experiments, as suggested by Reviewers 1 and 3. The new results have been added to Tables 2 and 3, demonstrating the consistently superior performance of our proposed method compared to existing methods. This reinforces the generalizability of our approach to several neural architectures.
>
> Q1: Thank you for the suggestion. We proofread the paper carefully to improve on Grammar and correct Typos. The main typos are listed as follows:
>
> Faderated > Federated; implementated > implemented; pairewise > pairwise; Mode > Model; solved the format issue of Algorithm 1 in Appendix A.
>
> We thank the Reviewer again for the useful comments. We have revised the manuscript according to the Reviewer’s suggestion and response to each comment provided in the Weakness section above. We hope that our rebuttal aligns with the reviewer’s expectations. The revised manuscript has proved the generalizability of the proposed method to more neural architects and the presentation has been improved according to Reviewers’ suggestions. We hope that the Reviewer can consider possibly giving a higher rating. Thanks.

---

### Author Response · Authors · 2023-11-19
**General Response**

We would like to express our sincere gratitude to all the reviewers for their invaluable comments, which have significantly contributed to enhancing the quality of our manuscript.

We appreciate the recognition from all reviewers regarding the substantial improvements our proposed method has made over existing methods in model merging. Special thanks to Reviewers 1, 2, and 4 for validating the solidity of our experimental setup, and to Reviewers 1, 2, and 3 for acknowledging the integrability of our scheme with existing methods. We are particularly encouraged by Reviewer 2's suggestion that our scheme could inspire novel algorithmic development.

The constructive feedback provided by the reviewers has been instrumental in refining our manuscript. Reviewers 1, 2, and 3 suggested additional experiments to bolster our results, while Reviewers 1, 3, and 4 expressed concerns about certain explanations and limitations. Valuable suggestions from Reviewers 2 and 4 regarding figure layout and manuscript structure have been duly considered.

We have addressed the comments comprehensively, as outlined below:

1.	Experimental Expansion:

* We have incorporated additional architectures, T5 and GPT2, into our experiments, as suggested by Reviewers 1 and 3. The new results have been added to Tables 2 and 4, demonstrating the consistently superior performance of our proposed method compared to existing methods. This reinforces the generalizability of our approach to several neural architectures.

* In response to Reviewer 2, we have added a discussion on coefficient search in Appendix D. The result is positive with further enhancement in performance. This addition highlights the potential for further improving performance of model evolution through integrated coefficient search for hyperparameters.

2.	Presentation and Explanation:

* Concepts, such as multi-task learning and the evolutionary algorithm for optimization, have been explained in more detail in the Introduction.

* Limitations related to data privacy setup, as suggested by Reviewer 1, have been explicitly addressed.

* Figure 1, Tables 2, 3 and 4 layouts have been improved to address concerns raised by Reviewer 2. We have also further enhanced the methodology and result description based on the feedback from Reviewer 3.

3.	Computational Cost and Accessibility:

* The computational cost of our proposed method has been detailed in Appendix C.5. Our results indicate that model evolution can be completed within an hour with moderate computing power, underscoring the practicality of our approach.

* The revised submission now includes source code to facilitate hands-on implementation, enhancing accessibility and convenience for researchers.

We sincerely thank the reviewers for their invaluable input, which has undeniably elevated the overall quality of our work. The meticulous examination of our manuscript has led to improvements that benefit both our research and its presentation. We believe that our comprehensive explanations and revisions align with the expectations of the reviewers and hope for favourable consideration for acceptance in ICLR.

In the rebuttal below, we address the weaknesses (W) and questions (Q) raised by each reviewer.

---

### Meta-Review · Area_Chair_7h48 · 2023-12-18

**Metareview:**

After careful review and consideration of the author's responses to the concerns raised by reviewers, the consensus remains that the paper needs more revision to be ready. The authors have shown effort in addressing concerns around the novelty and generality of their evolutionary algorithm approach to knowledge fusion, adding experiments with T5 and GPT-2 architectures. They also attempted to clarify their motivations and addressed the similarity in presentation with prior work (ICLR 2023 paper). However, key weaknesses highlighted by the reviewers still exist. The reviewers noted issues with the presentation and lack of strong novelty. Concerns about the considerable search cost and the absence of a clear theoretical framework for evolutionary-based model evolution being superior remains unaddressed. Furthermore, the manuscript's connection to prior work and the rationale for the evolutionary approach over simpler methods like coefficient search have not been convincingly articulated.

**Justification For Why Not Higher Score:**

Key weaknesses highlighted by the reviewers still exist. The reviewers noted issues with the presentation and lack of strong novelty. Concerns about the considerable search cost and the absence of a clear theoretical framework for evolutionary-based model evolution being superior remains unaddressed. Furthermore, the manuscript's connection to prior work and the rationale for the evolutionary approach over simpler methods like coefficient search have not been convincingly articulated.

**Justification For Why Not Lower Score:**

N/A

---

### Decision · Program_Chairs · 2024-01-16

Reject